# Clinical, Pathological, and Molecular Features of Breast Carcinoma Cutaneous Metastasis

**DOI:** 10.3390/cancers13215416

**Published:** 2021-10-28

**Authors:** Silvia González-Martínez, David Pizarro, Belén Pérez-Mies, Tamara Caniego-Casas, Giuseppe Curigliano, Javier Cortés, José Palacios

**Affiliations:** 1Clinical Researcher, Hospital Ramón y Cajal, 28034 Madrid, Spain; silviagonzalezmartinezbio@gmail.com; 2Fundación Contigo contra el Cáncer de la Mujer, 28010 Madrid, Spain; 3Department of Pathology, Hospital Ramón y Cajal, 28034 Madrid, Spain; david.pizarro@salud.madrid.org (D.P.); bperezm@salud.madrid.org (B.P.-M.); tamara880723@hotmail.com (T.C.-C.); 4Institute Ramón y Cajal for Health Research (IRYCIS), 28034 Madrid, Spain; 5CIBER-ONC, Instituto de Salud Carlos III, 28029 Madrid, Spain; 6Faculty of Medicine, University of Alcalá de Henares, Alcalá de Henares, 28801 Madrid, Spain; 7European Institute of Oncology, IRCCS, 20141 Milan, Italy; giuseppe.curigliano@ieo.it; 8Departament of Oncology and Hematology, University of Milan, 20122 Milan, Italy; 9Department of Medicine, Faculty of Biomedical and Health Sciences, Universidad Europea de Madrid, 28670 Madrid, Spain; 10International Breast Cancer Center (IBCC), Quironsalud Group, 08017 Barcelona, Spain; 11Medica Scientia Innovation Research, 08007 Barcelona, Spain; 12Medica Scientia Innovation Research, Ridgewood, NJ 07450, USA; 13Vall d’Hebron Institute of Oncology, 08035 Barcelona, Spain

**Keywords:** breast cancer, metastasis, skin, pathology, immunohistochemistry, mutation, CNV

## Abstract

**Simple Summary:**

Metastasis is the last stage in the development of cancer and usually results in mortality. Cutaneous metastases (CMs) account for 2% of all skin malignancies. Nearly 70% of CMs in women originate from breast cancer (BC). Since CM is usually associated with poor prognosis, the development of management strategies for these patients remains an important clinical challenge. Identifying molecular markers in primary BC that predict CMs and determining the molecular differences between primary tumors and their metastases is of great interest for designing new therapeutic approaches.

**Abstract:**

Cutaneous metastases (CMs) account for 2% of all skin malignancies, and nearly 70% of CMs in women originate from breast cancer (BC). CMs are usually associated with poor prognosis, are difficult to treat, and can pose diagnostic problems, such as in histopathological diagnosis when occurring long after development of the primary tumor. In addition, the molecular differences between the primary tumors and their CMs, and between CMs and metastases in other organs, are not well defined. Here, we review the main clinical, pathological, and molecular characteristics of breast cancer CMs. Identifying molecular markers in primary BC that predict CM and can be used to determine the molecular differences between primary tumors and their metastases is of great interest for the design of new therapeutic approaches.

## 1. Introduction

Breast cancer (BC) is the most prevalent neoplastic disease among women worldwide. In women, it represents around 30% of all new cancer cases and is the leading cause of cancer death [1,2].

Metastatic disease is the last stage in the development of cancer, and the outcome is usually fatal. It is a complex process in which the cells of a primary tumor propagate to distal organs by invading local tissues and blood and/or lymphatic vessels, followed by uncontrolled growth in these distal tissues [3,4]. BC cells predominantly metastasize to the lungs, bones, and brain [5], although the metastatic pattern varies among different subtypes. ER-positive tumors usually have less invasive and metastatic potential and tend to metastasize to the bone [6,7,8]. The incidence of brain metastases in HER2-positive BCs is high [9,10], and nearly 50% of patients with advanced disease die from central nervous system progression [11]. TNBCs frequently metastasize to visceral organs; they show a high tropism for the lungs but also a high rate of metastasis to the central nervous system [12].

Cutaneous metastases (CMs) account for only 2% of all skin cancers [13,14]. In women, BC is the tumor most likely to metastasize to the skin [13,15,16,17]. Notably, 69% of the CMs in women observed by dermatologists originate from BC [17]. These CMs result from lymphatic embolization and hematogenous or contiguous dissemination [18] and are present in around 24% of patients with metastatic BC [13,14,15,16,17]. They can appear locally (86%) or distantly (14%). Since CM is usually associated with poor prognosis [19], the development of management strategies for these patients remains an important clinical challenge. Identifying molecular markers in primary BC that predict CM and could be used to identify CMs would be of great clinical interest. In this narrative review, we discuss the main clinical, pathological, and molecular features of CM from the breast.

## 2. Clinical and Pathological Features of Breast Cancer Cutaneous Metastasis

### 2.1. Epidemiology, Incidence, and Prognosis of Breast Cancer Cutaneous Metastasis

The frequency of CM varies according to the molecular subtype of BC. In their study, Lefebvre et al. [20] included 28 cases of CM in which the BC had the following phenotypes: 61% HR-positive/HER2-negative, 25% TN, 7% HER2-positive, and 7% unknown. Yates et al. [21] included 19 cases of CM from BC, which were 47% ER-positive/HER2-negative, 26% TN, 10% ER-positive/HER2-positive, and 16% unknown. The distributions of cases among the different molecular phenotypes in the studies of Kong et al. [22] and Luna et al. [23] differ from those in the remainder of the series in the number of cases HER2-positive and also the series of Luna et al. [23] in the number of cases TN; Kong et al. [22] included 125 women with skin and/or soft tissue metastasis, in which 42.4% were HR-positive, 34.4% were HER2-positive, and 23.2% were TN; and Luna et al. [23] included 26 cases, 27% of which were HR-positive/HER2-negative, 27% HER2-positive, 39% TN, and 7% unknown (Table 1).

Although there are differences among series, a finding common to all of them is the relative overrepresentation of TNBC, since its frequency in the general population of BCs is around 15%, but it is between 23% and 39% in BCs with CM.

The median age of women who have BC with CM is 74 years [13,15,16,17,24,25]. As previously mentioned, CM usually occurs late in the disease, in the later stages of the cancer course. Brownstein et al. [26] analyzed 724 cases of metastatic cancers with skin metastases and observed that signs of skin metastasis were only present in 3% of metastatic BC cases. Kong et al. [22] observed that more than half of the patients (56.8%) had more than one visceral metastasis at the time of CM diagnosis: bone in 41.6%, lung in 36%, liver in 13.6%, and brain in 1.6% of patients. At the time of diagnosis, the primary tumor was already widespread, and curative treatment was not available.

### 2.2. Clinical Presentation of Breast Cancer Cutaneous Metastasis

The location of CMs in BC patients is not completely random but occurs more frequently in the vicinity of the primary tumor, perhaps by direct extension of the underlying tumor or by lymphatic spread [16]. CMs from BC tend to develop in the chest wall, although they can also develop in the abdomen, extremities, head, or neck [13,16,17,27]. In addition to being able to develop in different locations, cutaneous breast metastasis can manifest with different patterns.

The most common clinical presentation of skin metastases is in the form of nodules, with an incidence of 80% [17]. They range in size from 1 to 3 cm and may present as firm, solitary, or multiple papules that are located in the dermis or subcutaneous tissue. They are often flesh colored but could be colored brown, bluish black, pink, or red–brown [28,29,30]. The nodules are usually asymptomatic, although they can also become ulcerated and infected [31] (Figure 1a).

The telangiectatic pattern may appear as an erythematous patch with prominent telangiectasias or as a lymphangioma circumscriptum-like pseudovesicular lesion [32]. It is characterized by papules, plaques, or purpuric nodules, and it is often accompanied by pruritus [33,34,35]. Telangiectatic presentation occurs with an incidence of 8–11% among patients [15,17].

The erysipeloides or erysipelatoides pattern is also known as inflammatory metastatic carcinoma [30,36]. This pattern shows erysipelas-like lesions, with sharply demarcated erythematous patches and plaques affecting the breast and surrounding skin. Erysipeloides occurs with an incidence of 3–6.3% among patients [15,17] (Figure 1b).

Carcinoma en cuirasse, also named scirrhous carcinoma, appears as scattered and firm erythematous and indurated plaques on the chest wall [30,37,38]. Carcinoma en cuirasse occurs with an incidence of 3–4% among patients [15,17].

Neoplastic alopecia presents as circular indurated areas of alopecia on the scalp due to the hematogenous spread of BC [39]. The areas of alopecia are typically painless, nonpruritic, and well demarcated, and they appear as oval plaques, often displaying a red–pink tone and a smooth surface [26]. The infiltrative nature of such alopecia may be unapparent or only minimally apparent [40]. This type occurs with an incidence of 2–12% among patients [15].

### 2.3. Histopathology of Breast Cancer Cutaneous Metastasis

The main BC histological type that produces CMs are an invasive carcinoma of no special type (NST). In the series reported by Mayer et al. [41], 95.2% of CM cases resulted from BCNST, followed by invasive lobular carcinoma (ILC) (2.4%) and others (2.4%).

The pattern of infiltration of dermal tissue influences the clinical pattern of presentation. Histologically, nodules are composed of atypical neoplastic cells arranged in small nests and cords surrounded by fibrosis, usually in a single-file line within the collagen bundles of the dermis [28,31]. In the telangiectatic pattern, aggregates of atypical neoplastic cells and erythrocytes are present within dilated vessels of the papillary and/or reticular dermis (Figure 2) [31]. In the erysipeloid pattern, the metastatic tumor cells are tightly crowded within dilated superficial and deep lymphatic vessels, and a slight perivascular infiltrate of lymphocytes and plasma cells is usually present [15,31,32]. Carcinoma en cuirasse forms through dense fibrosis with few neoplastic cells, sometimes exhibiting a characteristic single-file pattern between the collagen bundles of the dermis [15,31]. Finally, neoplastic alopecia is characterized by the presence of small tumor cells arranged in cords, with single cells destroying hair follicles and inducing fibroplasia [42,43,44,45].

For CMs, it is recommended that immunohistochemical analysis detection of ER, PR, and HER2 (Figure 3) be repeated to evaluate whether the immune profile of the primary tumor has changed, since a change can influence the efficacy of future treatment.

#### Differential Diagnosis of CM of Mammary Origin

Histologically, the primary characteristics of the neoplasia can be observed in CM, but CMs are often very poorly differentiated variants of the original tumor, and a microscopic study may be insufficient to arrive at the etiological diagnosis [46]. Certain immunochemistry markers (IHC) are used in the investigation of CM origin in cases where a primary tumor has not yet been diagnosed at the time of cutaneous biopsy. Against CM of unknown origin, it is often useful to perform IHC for cytokeratin 7 (CK7), cytokeratin 20 (CK20), antithyroid transcription factor 1 (TTF-1), S100 protein (S100), and HRs. The distinction of the skin lesion as resulting from BC metastasis has great clinical importance since it can completely modify the application of treatment guidelines. To date, there is no totally specific marker for confirming breast origin. Despite not being specific markers to define the mammary origin, a CK7 +, CK20 −, ER + and PR + algorithm is sometimes used [25,47]. Additionally, there are multiple antibodies that can help to distinguish it from primary skin tumors or metastases of other origins.

The difficulty in recognizing the tumor as being of mammary origin mainly depends on the profile of primitive BC. In luminal tumors, which present intense expression of HRs, and in HER2 tumors, the diagnosis can be made more easily, since these tumors present the characteristic immunophenotype of mammary origin with positivity for GCDFP (gross cystic disease factor protein), mammaglobin, GATA3, HRs, or HER2, if applicable.

TN cases are the most difficult group to diagnose. This subgroup of tumors presents much lower positivity for typical breast markers, such as mammaglobin (17–24% positivity), GCDFP-15 (0–5%), and GATA3, which show a lower expression than that identified in luminal carcinomas or HER2 [48].

GCDFP and mammaglobin are frequently used as immunohistochemical markers of mammary origin. Mammaglobin is more sensitive and GCDFP more specific, but both markers are expressed in most luminal and HER2 tumors. Staining is cytoplasmic and heterogeneous, so the entire stained slide should be carefully examined. However, these two markers are much less sensitive for TN tumors (< 35% and 16%, respectively) and therefore have less diagnostic utility for them. Cutaneous adnexal tumors can also express these two markers, which must be taken into account for differential diagnosis [49].

GATA3 is a transcription factor that plays a role in the differentiation of many tissues, including the mammary luminal epithelium. In surgical pathology, GATA3 is a sensitive marker of BCs since GATA3 is involved in ER signaling, but it is also expressed in urothelial, trophoblastic, salivary, and pancreatic tumors, among others [49]. GATA3 is more sensitive than other markers, such as mammaglobin and GCDFP15, since its positivity in luminal tumors is around 90%, but in TN tumors, the percentage of positive tumors is lower. Its sensitivity is much higher for luminal tumors than for TN tumors [49].

SOX10 is a transcription factor that plays a role in the differentiation and survival of neural crest cells. Under normal conditions, SOX10 is expressed in normal salivary gland tissue, in bronchial cells, and in myoepithelial cells of the breast. In surgical pathology, SOX10 has mainly been used as a marker for melanoma and nerve sheath tumors. In breast, the tumor expression of SOX10 has been related to the activation of progenitor cells and the epithelial–mesenchymal transition. SOX10 positivity has been described in 66–74% of TN and metaplastic BCs but only 5% of non-TN carcinomas. This marker can be useful even in the absence of GATA3 expression [49].

It is important to note that both ER-positive and ER-negative tumors can show intense S100 expression, which implies a differential diagnosis of melanoma.

Lastly, the androgen receptor plays a role in the normal development and proliferation of breast cells. It is expressed more frequently than ER and PR, but like GATA3, its expression is closely linked to ER expression, with a positivity between 25% and 35% in TN carcinomas [49].

SOX10, androgen receptor, and GATA3 expression studies have recently been carried out in TNBC, both in primary tumors and in metastatic carcinomas [45] (Figure 4 and Figure 5). In the review of Tozbikian et al. [49], the most sensitive of the three markers is GATA3 (82% positivity), and the least sensitive is the androgen receptor. SOX10, in turn, remains the most stable when comparing the expression between primary and metastatic carcinoma, and in addition, it is capable of allowing the detection of cases in which there is no expression of GATA3.

Another important differential diagnosis of CM is primary epithelial cutaneous tumors, mainly when metastasis develops a long time after the primary tumor. The distinction between primary adnexal carcinoma (PAC) and metastatic BC to the skin can be quite challenging, despite adequate clinical history [50,51]. The differential diagnosis with primary cutaneous apocrine carcinoma, a rare tumor that shares morphologic characteristics with BC is especially defiant. For this subtype, GATA 3 has been proven useful in confirming breast origin, but this antibody can be also expressed in other primary skin tumors [50]. CD117 (is expressed by sweat gland secretory cells as well as a wide variety of adnexal tumors, so it can be useful in differentiating tumors with apocrine/eccrine differentiation from metastatic carcinomas to the skin with a high specificity, albeit low sensitivity. D2-40 (podoplanin) has been reported to highlight not only a variety of vascular tumors but is a useful tool in distinguishing PAC and other cutaneous tumors from metastatic adenocarcinoma to the skin. P63 expression has also been found to be helpful in distinguishing primary adnexal tumors from metastatic adenocarcinoma to the skin, as it is highly expressed in the basal/myoepithelial cells of epithelial tissues, such as in the eccrine/apocrine sweat glands. Therefore, the use of these three markers in a panel may serve as a valuable tool in daily clinical practice when faced with this dilemma [51]. However, it should be taken into account that some TNBC can express one or more of these proteins.

### 2.4. Prognosis and Treatment of Breast Cancer Cutaneous Metastasis

The poor prognosis conferred by the presence of CMs in women with BC is highlighted by the short median survival of these patients. There are few differences in prognosis regarding whether the lesions are single or multiple, with a mortality exceeding 70% in the first year after the diagnosis [13,52].

When BC spreads to the skin, it cannot be cured. The purpose of treatment is to relieve symptoms, improve quality of life, and slow the growth of the cancer. The CMs are systemically treated based on the molecular type (HR-positive, HER2-positive, or TN). A HR-positive tumor can be treated with a variety of endocrine-based strategies. Chemotherapy is the preferred option for CMs that are HR-negative and/or rapidly progressing. HER2-directed therapy should be applied to HER2-positive tumors, depending on the extent of disease. For the local control of skin metastases, external beam radiation therapy is an option for palliation but would not usually be used in previously irradiated areas due to the issue of cumulative dose [17].

Debridement may be advised when lesions are bleeding. Other options are possible and can be helpful, such as imiquimod, showing good results in localized lesions; trastuzumab, in tumors that are HER2-positive; systemic chemotherapy; radiation therapy; and immunotherapy [53,54]. Surgical excision may improve the quality of life of patients [54,55].

In a number of cases, it has been highlighted that cryotherapy and topical immunomodulators may be useful for the management of CM of BC, possibly by eliciting improved responses to traditional systemic chemotherapy, as may emerging therapies including targeted agents and immune-checkpoint blockers. Trials evaluating combination therapy in metastatic melanoma (NCT03276832) are ongoing. Clinical and correlative studies to develop a therapeutic protocol for CM of BC based on these observations are underway [56].

Another option for the treatment of CM in BC is electrochemotherapy (ECT). This is an effective treatment for cutaneous BC lesions that have proven refractory to standard therapies. As smaller lesions were found to be more responsive, Bourke et al. [36] suggested that ECT should be considered as an early treatment modality, within multimodal treatment strategies.

Elucidating the molecular landscape of CMs could help in selecting new targeted therapies for this ominous manifestation of BC.

A potential diagnostic and treatment algorithm for CM is presented in the Figure 6.

## 3. Molecular Landscape of BC Metastasis

### 3.1. Overview of Molecular Alterations in Metastases of Mammary Origin

Metastases are associated with the acquisition of additional driver mutations compared with the primary tumor [21]. Different studies show that the number of somatic mutations in BC metastasis was significantly higher than that in primary cancers [21,57,58,59,60]. The type of mutation acquired during the metastatic process will probably depend on both the intrinsic molecular subtype of BC and the metastatic site.

Rinaldi et al. [61], who explored a set of 11,616 breast tumors, including 5034 non-paired metastases, showed a significant enrichment for *ESR1* mutations in metastasis (18.3% in metastases vs. 2.2% in local disease). Additionally, they observed a metastatic enrichment of previously unreported, lower-prevalence *ESR1* mutations in the ligand-binding domain, implying that these mutations may also be functional. The type of *ESR1* mutation differed according to the histological type (ductal vs. lobular) and the metastatic site. Other alterations enriched across all the metastases include the loss of function of *CDKN1B* and mutations in the transcription factor *CTCF*, *NF1*, and *KRAS.* Mutations enriched at specific metastatic sites affected genes that are important in the development of primary tumors in that site, suggesting the local adaptation of BC metastasis. They included 118 cases of CM and demonstrated that only changes in *NOTCH1* were specific for skin metastases. Other examples included *PTEN* and *ASXL1* in brain metastases and *KRAS*, *KEAP1*, *STK11*, and *EGFR* mutations in lung metastases.

Lefebvre et al. [20], in their study of 216 metastatic breast tumors, also observed more *ESR1* mutations in metastases than in primary tumors, as well as more mutations in *RB1*. In this study, other genes that were more frequently observed to be mutated in metastasis from BC as compared with early BC were *FSIP2*, *FRAS1*, *OSBPL3*, *EDC4*, *PALB2*, *IGFN1*, and *AGRN*.

Focusing on studies that analyzed specific metastases, Tian et al. [62], using the MSKCC Dataset and POTUAM dataset, studied cases of hepatic metastases and unpaired primary breast tumors. They found that the driver genes for liver metastasis (LM) were *ESR1*, *AKT1*, *ERBB2*, and *FGFR4* and that LM matched three prominent mutation signatures: APOBEC cytidine deaminase, ultraviolet exposure, and defective DNA mismatch repair. Huang et al. [63] studied cases of brain metastases and unpaired primary breast tumors and observed that the metastases were enriched for genomic alterations in *TP53*, *ERBB2*, *RAD21*, *NF1*, *BRCA1*, and *ESR1*. They also demonstrated significantly increased levels of immune-checkpoint-inhibitor biomarkers, microsatellite instability, and *CD274* amplification in the metastasis cohort.

#### Differences in Mutational Profile between Primary Tumors and Their Paired Metastases

Some studies have compared mutations between paired cases of primary breast tumors and their metastasis. Schijver et al. [57] included 17 paired ER-negative/HER2-positive and TN cases and found that a large subset of non-synonymous somatic mutations (45%) were shared between primary tumors and paired metastases. However, mutations restricted to a given primary tumor or its metastasis, the acquisition of a loss of wild-type allele heterozygosity, and clonal gene changes affected by somatic mutations, such as in *TP53* and *RB1*, were observed in the progression of primary tumors to metastasis. Roy-Chowdhuri et al. [59] analyzed 61 matched cases and observed 77% concordant mutations between primaries and metastases and 23% mutations only in metastases, including *TP53*, *PIK3CA*, *KRAS*, *PTEN*, *BRAF*, and *AKT1.* However, their group of patients focused primarily on the ER-positive subtype and included only 2% that were ER-negative/HER2-positive breast tumors and 25% that were TN breast tumors. Other studies have also shown that *TP53* and *PIK3CA* are the genes most frequently mutated in metastases [21,64]. Paul et al. [58] studied 28 matched cases and observed seven genes that were preferentially mutated in metastases—*MYLK*, *PEAK1*, *SLC2A4RG*, *EVC2*, *XIRP2*, *PALB2*, and *ESR1*—five of which are not significantly mutated in any type of human primary cancer. Kjällquist et al. [60], who included 30 matched cases, showed the enrichment of mutations in *AKAP* genes in metastatic BCs, suggesting the involvement of *AKAPs* in the metastatic process. Furthermore, Van Geelen et al. [65] studied 76 paired cases and found alterations in *AKT1*, *BRCA2*, *CHEK2*, *ESR1*, *FGFR1*, *KMT2C*, *NCOR1*, *PIK3CA*, and *TSC2* that were significantly enriched in metastases when compared with paired primary tumors. On the other hand, they observed agreement in both types of samples in cases of *TP53* and *ERBB2* amplification. Paul et al. [58] included 28 paired cases in their study and found preferential mutations or copy number variations (CNVs) to be altered in metastases compared with paired primary tumors. The genes preferentially mutated in metastases were *MYLK*, *PEAK1*, *SLC2A4RG*, *EVC2*, *XIRP2*, *PALB2*, and *ESR1* (five of which are not significantly mutated in any type of human primary cancer). The most relevant CNVs were a loss of *STK11* and *CDKN2A/B* and gain of *PTK6* and the membrane-bound PR *PAQR8. PAQR8* gain and *ESR1* mutations were mutually exclusive, suggesting a role in treatment resistance.

A recent specific retrospective study of 94 paired cases of lobular carcinoma demonstrated mutations in *AKT1*, *ARID1A*, *ESR1*, *ERBB2*, or *NF1* only in 22% of patients with metastasis; *PTEN* and *NF1* deletion and *CYP19A1* amplification were seen in 19% of patients [66].

### 3.2. Molecular Landscape of Breast Cancer Cutaneous Metastases

Some of the studies that have analyzed mutations in metastatic lesions and mutations in their respective primary tumors specify additional mutations observed in skin metastases not seen in the primary tumors [21,58,60]. Data for these are presented in Table 2.

Referring to copy number alterations, Moelans et al. [67], who included 22 cases of primary and skin metastases from a total of 55 primary BC samples and their corresponding distant metastases, showed a high frequency of CNVs in BC metastases compared with primary tumors for a few genes (*ADAM9*, *IKBKB*, *PRDM14*, *CCND1*, *MED1*, *ERBB2*, *CDC6*, *C11ORF30*, *CDH1*, *TRAF4*, *CPD*, *CDC6*, *MAPT*, and *CCNE1)*. These genes had been implicated in different pathways, including the development of therapy resistance. Focusing only on paired skin cases, they observed more gains of the *ESR1*, *MTDH*, *MED1*, and *ERBB2* genes in metastasis samples than in the primary tumors and more *CDH1* losses in the metastasis compared with the primary tumor. Curigliano et al. [68] identified a molecular signature on skin metastasis. This molecular signature was similar but not identical to that described for the basal-like subtype, including high expression levels of several cytokeratins, Aurora kinase A, Cyclin E, and others.

### 3.3. Comparison of Mutational Profile between Cutaneous and Hepatic Metastases

In order to further explore the possible specific molecular alterations of CMs in comparison with other visceral metastases, we analyzed the mutational differences between CMs and LMs of BC. We chose LMs because they are the metastases most frequently found in the analyzed series, probably because they are easier to biopsy. We included 58 sequenced cases of skin metastasis from five series [20,21,57,58,69] and 87 cases of LM from two series [20,21]. We could not access the data of the largest series of metastases reported by Rinaldi et al. [61].

First, we analyzed whether there were differences in the immunophenotypes of BC according to the metastatic site. The percentages of each phenotype of BC with skin metastasis with respect to those with LMs is presented in Table 3.

Although the differences in the immunophenotype were not statistically significant, which could be due to the relative low number of cases, we observed a higher proportion of the TN phenotype in tumors with CMs than in those with LMs.

The mutations more frequently found in CMs and LMs are shown in Figure 7.

For this dataset, we evaluated statistically significant differences, considering only those genes that appeared mutated in at least four samples. We observed differences in six genes: *RUNX1*, *ROBO1*, *MLL2*, *DYNC2H1*, and *CHD1* were mutated in 7% of CMs but no LMs (*p*-value = 0.025), and *TP53* was mutated in 50% of skin metastases and 31% of LMs (*p*-value = 0.036) (Figure 8). This analysis has certain limitations, since the data included come from different sequencing platforms and different numbers of genes have been sequenced according to the study. Lefebvre et al. [20] and Paul et al. [58] analyzed the whole exome. On the other hand, Yates et al. [21], Schrijver et al. [57], and Muller et al. [69] sequenced a specific panel of genes. Additionally, further studies with larger series are required to check whether the discrepant results were related to differences in the molecular landscapes of the primary tumors or represented specific events in CMs [70].

According to the alterations observed in CM and LM, the biological mechanisms underlying their development can be hypothesized. Figure 9a shows the number of altered genes of each molecular pathway and the number of samples in which genes of the pathway are mutated in CM. Figure 9b presents the same information referring to LM. In both CM and LM, the pathway most affected according to the mutations found is PI3K, followed by TP53 and RTK-KRAS, with the frequencies 58.6% vs. 46.5%, 50% vs. 36% and 46.5% vs. 30.2%, respectively.

## 4. Possible Future Directions

To improve the knowledge of the biology of BCM, it would be necessary to carry out more sequencing studies with a greater number of cases from patients with CM of mammary origin. More paired cases of primaries with their corresponding CM should be studied with a broad panel of genes, including all those most frequently involved in BC, to find the additional mutations in CM that lead to their development, to identify not only metastasis mechanisms but also potential therapeutic targets. Another possible future projection in this area would be using whole exome sequencing to study the primary BC tumors of patients who have developed CM, attempting to find a specific mutational profile in such patients in comparison with tumors that developed metastases in other sites. Studies of this type could have clinical impact since knowing the pathways that lead to the development of skin metastases or knowing a specific genetic profile in the primary tumor of the patients who develop this type of metastasis would help to act against specific targets or help to predict and prevent tumor development to the skin.

## 5. Conclusions

CM is a factor for poor prognosis of BC, and new therapeutic approaches are needed to improve patient survival. A better understanding of the pathological and molecular characteristics of the primary tumor and their CMs will help in achieving this objective.

## Figures and Tables

**Figure 1 cancers-13-05416-f001:**
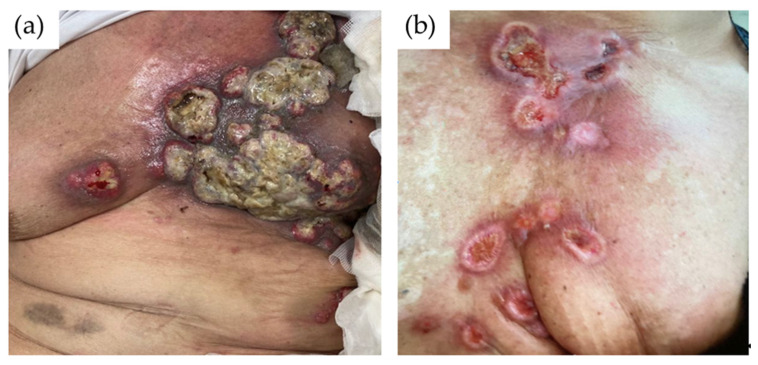
(**a**) Cutaneous metastases in form of ulcerated and infected nodules. (**b**) Cutaneous metastasis in the form of sharply demarcated erythematous patches and plaques affecting the breast and surrounding skin.

**Figure 2 cancers-13-05416-f002:**
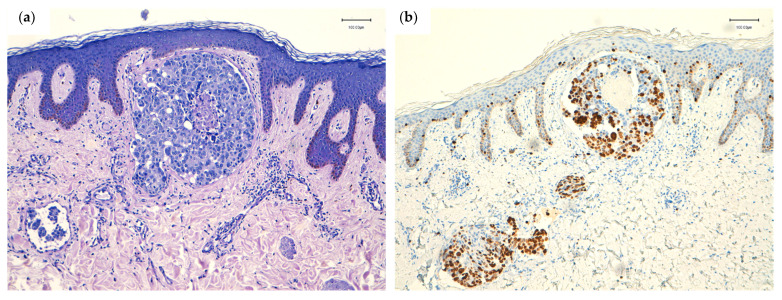
(**a**) Lymphovascular permeation in dermal lymphatics. H&E; original magnification, 100×. (**b**) High proliferation index in tumor emboli (Ki-67; Clon Mib-1, Agilent). Original magnification, 100×.

**Figure 3 cancers-13-05416-f003:**
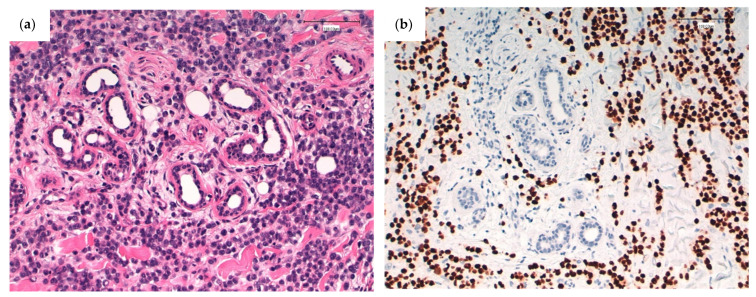
(**a**) Cutaneous metastasis from a lobular carcinoma. Discohesive, monomorphic cells surrounding a cutaneous appendage. H&E; original magnification, 200×. (**b**) High expression of estrogen receptor (Clon EP1, Agilent). Original magnification, 200×.

**Figure 4 cancers-13-05416-f004:**
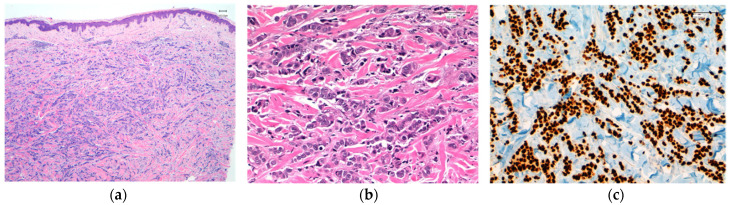
(**a**) Infiltrating trabecular pattern in a cutaneous nodular metastasis from a triple-negative breast carcinoma of no special type. Original magnification, 40 ×. (**b**) Tumor cells dissecting dermal collagen fibers. Original magnification, 200×. (**c**) Intense expression of GATA3 (Clon L50-823, Roche). Original magnification, 200×.

**Figure 5 cancers-13-05416-f005:**
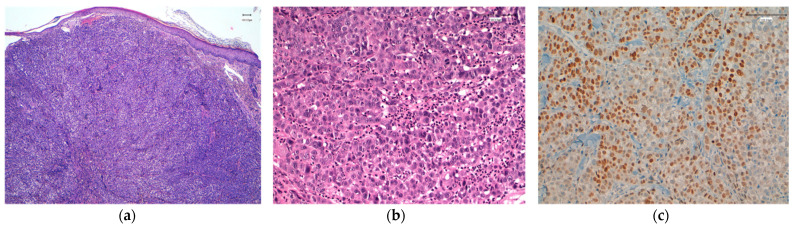
(**a**) Solid pattern of infiltration in a cutaneous nodular metastasis from a triple-negative breast carcinoma of no special type. Original magnification, 40 ×. (**b**) Tumor cells forming solid nests in the dermis. Original magnification, 200×. (**c**) Intense expression of SOX10 (Clon SP267, Roche). Original magnification, 200×.

**Figure 6 cancers-13-05416-f006:**
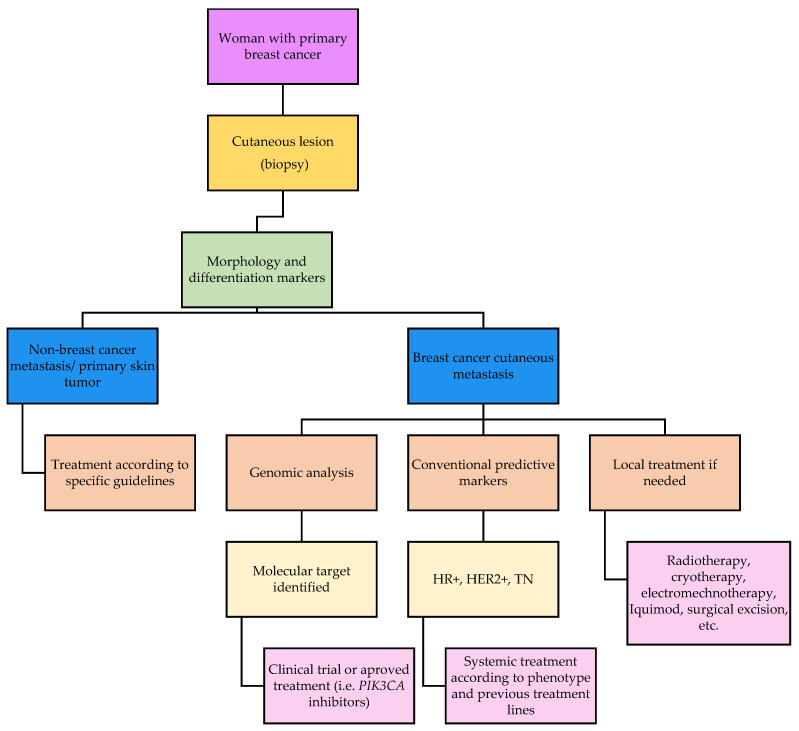
Diagnostic and treatment algorithm for cutaneous metastasis.

**Figure 7 cancers-13-05416-f007:**
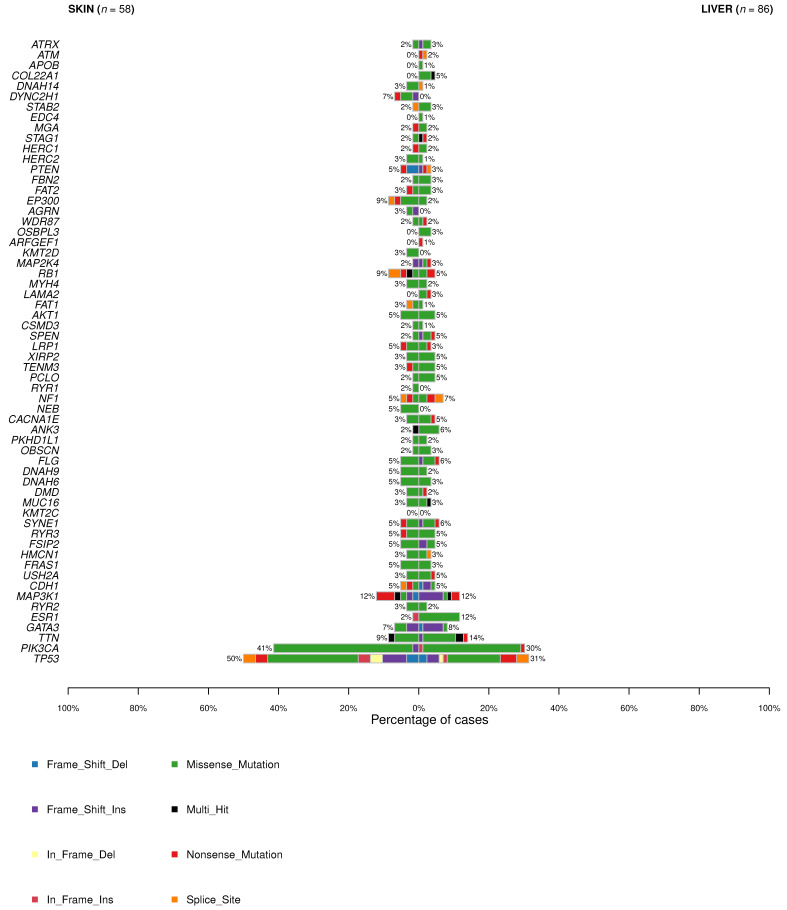
Type and frequency of mutations in cutaneous and liver metastases according to gene.

**Figure 8 cancers-13-05416-f008:**
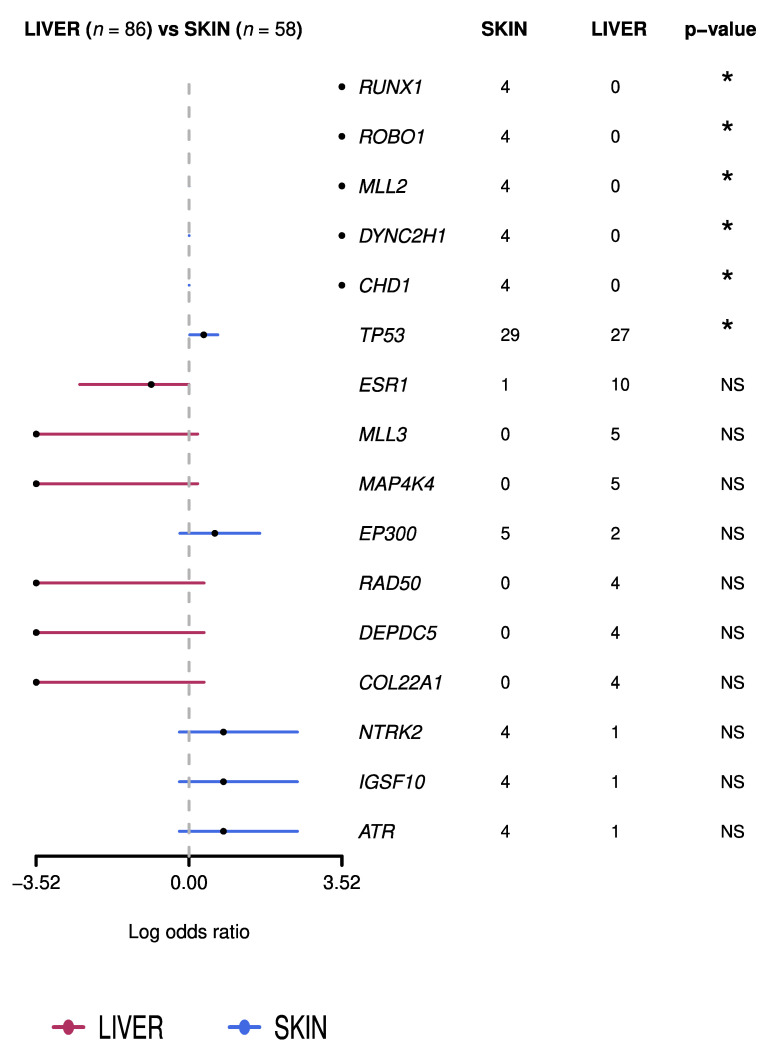
Genes differentially mutated in liver and skin metastases (* statistically significant differences). The red lines reflect a higher frequency of mutation in genes in the liver metastases with respect to the skin metastases, with no statistically significant differences, and the blue lines reflect a higher frequency of mutation of genes in the skin metastases with respect to the liver metastases, with no statistically significant differences.

**Figure 9 cancers-13-05416-f009:**
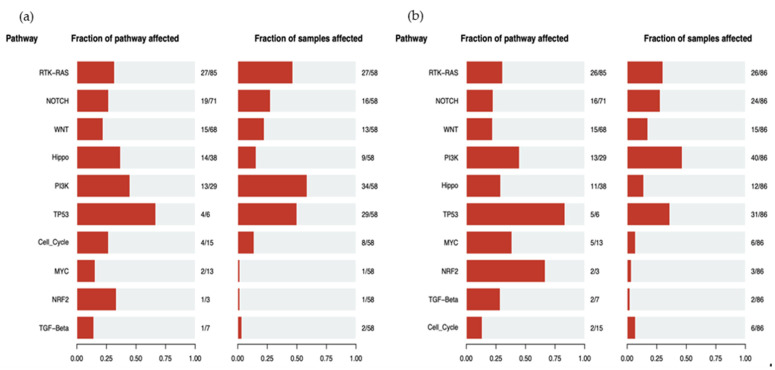
(**a**) Altered pathways according to the number of mutated genes and according to the number of samples affected in cutaneous metastasis. (**b**) Altered pathways according to the number of mutated genes and according to the number of samples affected in liver metastasis.

**Table 1 cancers-13-05416-t001:** Distribution of molecular phenotypes in the cases of BC with MC of each series.

References	N	HR-Positive/HER2-Negative N (%)	HER2-Positive N (%)	TN N (%)	Unknown N (%)
Lefebvre et al. [20]	28	17 (61)	2 (7)	7 (25)	2 (7)
Yates et al. [21]	19	9 (47)	2 (10)	5 (26)	3 (16)
Kong et al. [22]	125	53 (42.4)	43 (34.4)	29 (23.2)	
Luna et al. [23]	26	7 (27)	7 (27)	10 (39)	2 (7)

**Table 2 cancers-13-05416-t002:** Additional mutations in skin metastases vs. primary breast tumors in paired cases along with data source.

References	Total Paired Cases N	Paired Cases of Skin Metastases N	Additional Molecular Alterations in Skin Metastases Not Found in Primary Breast Tumors
Schrijver et al. [57]	17	8	33 mutations (*ATR*, *BRCA1*, *SMAD4*, *CDH1*, *ARID1A*, *ERBB2*, *IDH1*, *PIK3R1*, *RB1*, and others)
Yates et al. [21]	Cohort 1: 7	2	4 molecular alterations (amplification of *FGFR1*/structural variant of *TP53*, indel of *RB1*/amplification of *TERC*)
	Cohort 2: 51	4	9 mutations (*JAK2*, *NF1*, *TP53*, *AKT1*, and *ARID1A*)
Paul et al. [58]	28	1	54 mutations (*PIK3CA*, *TP53*, and others)

**Table 3 cancers-13-05416-t003:** Number of cases of metastasis in skin and liver grouped by phenotype.

Location	Luminal N (%)	Luminal HER2-Positive N (%)	HER2-Positive N (%)	TN N (%)	ND N (%)
SKIN	29 (50)	2 (3.4)	6 (10.3)	15 (25.9)	6 (10.3)
LIVER	59 (68.6)	4 (4.6)	9 (10.5)	11 (12.8)	3 (3.5)

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
