# Peer review of "Clinical, Pathological, and Molecular Features of Breast Carcinoma Cutaneous Metastasis"

_cancers, 2021, doi:10.3390/cancers13215416_

Round 1
Reviewer 1 Report
González-Martínez et al report an important review on the cutaneous metastases from breast cancer. This is especially clinically relevant given the difficulties and complexities in managing this patient population, albeit a rare subgroup.
However, the review may benefit from greater discussion on the clinical management and treatment of cutaneous metastases. The clinical and pathological features have been well described, although the literature remains sparse with regards to specific characterization of cutaneous metastases. Similarly, although there may be limited data with regards to treatment and management, providing a potential diagnostic and treatment algorithm for cutaneous metastases may unify the different sections and increase the clinical utility of the paper.
The evaluation of mutational profiles comparing cutaneous and hepatic metastases is interesting, albeit exploratory in nature. The limitations of this analysis should be more clearly described – in particular, the pooled datasets which may have had significant differences in sequencing methodologies, especially the breadth of NGS panel used. In addition, greater discussion into the potential biologic mechanisms underlying the development of cutaneous metastases which might be hypothesized from the genes which were detected is warranted.
Finally, a section on future directions would add to the manuscript. This includes potential avenues for investigation for translational studies (e.g. deeper multi-omic sequencing of larger cohorts of patients with cutaneous metastases) or clinical studies (e.g. studies of local therapies for cutaneous metastases).
Reviewer 2 Report
In the present review, the author summarizes clinical, pathological, and molecular characteristics of breast cancer cutaneous metastasis. The review is complete and detailed, well structured and well written.
I have some minor points that I think the authors need to address.
- The figure numbers (Fig 2 a,b; Fig 3 a,b,c; Fig 4 b,c) are of poor quality and difficult to read. It need to be changed.
- The authors use different datasets to analyze mutational differences between skin metastases and liver metastases. Do the authors have their own data on the mutational profile of skin metastases and liver metastases? Or just skin metastases? It would be interesting to include such data in this review.
Reviewer 3 Report
I thank the authors and the editorial board for having had the opportunity to review this interesting article. In the proposed work, the authors provide a review on the clinical, pathological and molecular characteristics of skin metastases from breast cancer. The proposed work focuses on a topic of great clinical interest even if it suffers from the limitations of a narrative review. Breast cancer represents a clinical and diagnostic challenge all over the world and thanks to screening programs and new therapeutic approaches, an increase in survival and recovery has been observed in recent years. As stated by the authors, metastases represent an important poor prognostic factor and raise questions about treatment. Overall the work is interesting and well written, however some points should be addressed in depth and the paper cannot be accepted in its current form requiring major revisions
- In the paragraph on the clinical presentation of the macroscopic photos they would add value to the paper
- Patients with breast cancer often have long survivals and the histopathological features of diifferential diagnosis with primary tumors (e.g. adnexal tumors) should be systematically covered.
- A systematic review of the literature would represent a crucial improvement for the work
Round 2
Reviewer 1 Report
The authors have adequately addressed my comments.
Reviewer 3 Report
The authors responded effectively to the proposed remarks adding value to their work which can be accepted in its current form